# Radiation Induces Bone Microenvironment Disruption by Activating the STING-TBK1 Pathway

**DOI:** 10.3390/medicina59071316

**Published:** 2023-07-16

**Authors:** Yuyang Wang, Li Ren, Linshan Xu, Jianping Wang, Jianglong Zhai, Guoying Zhu

**Affiliations:** 1Institute of Radiation Medicine, Fudan University, 2094 Xietu Road, Shanghai 200032, China; 18211140008@fudan.edu.cn (Y.W.); 20211140001@fudan.edu.cn (L.R.); 19211140005@fudan.edu.cn (L.X.); jianpingwang@fudan.edu.cn (J.W.); jlzhai@fudan.edu.cn (J.Z.); 2Shanghai Municipal Center for Disease Control & Prevention, Shanghai 200051, China

**Keywords:** radiation, bone microenvironment, osteocyte, stimulator of interferon genes (STING), senescence-associated secretory phenotype (SASP), osteoclastogenesis

## Abstract

*Background and Objectives*: Damage to normal bone tissue following therapeutic irradiation (IR) represents a significant concern, as IR-induced bone microenvironment disruption can cause bone loss and create a more favorable environment for tumor metastases. The aim of the present study was to explore the cellular regulatory mechanism of IR-induced bone microenvironment disruption to effectively prevent radiotherapy-associated adverse effects in the future. *Materials and Methods*: In this study, a mouse model of local IR was established via local irradiation of the left hind limb of BALB/c mice with 12 Gy X-rays, and an in vitro osteocyte (OCY) model was established by exposing osteocyte-like MLO-Y4 cells to 2, 4, and 8 Gy irradiation to analyze multicellular biological injuries and cellular senescence. Small interfering RNA (siRNA) transfection at the cellular level and a selective antagonist intervention C-176 at the animal level were used to explore the potential role of the stimulator of interferon genes (STING) on IR-induced bone microenvironment disruption. *Results*: The results showed that 12 Gy local IR induces multicellular dysfunction, manifested as ascension of OCYs exfoliation, activation of osteoclastogenesis, degeneration of osteogenesis and fate conversion of adipogenesis, as well as cellular senescence and altered senescence-associated secretory phenotype (SASP) secretion. Furthermore, the expression of STING was significantly elevated, both in the primary OCYs harvested from locally irradiated mice and in vitro irradiated MLO-Y4 cells, accompanied by the markedly upregulated levels of phosphorylated TANK-binding kinase 1 (P-TBK1), RANKL and sclerostin (SOST). STING-siRNA transfection in vitro restored IR-induced upregulated protein expression of P-TBK1 and RANKL, as well as the mRNA expression levels of inflammatory cytokines, such as IL-1α, IL-6 and NF-κB, accompanied by the alleviation of excessive osteoclastogenesis. Finally, administration of the STING inhibitor C-176 mitigated IR-induced activation of osteoclastogenesis and restraint of osteogenesis, ameliorating the IR-induced biological damage of OCYs, consistent with the inhibition of P-TBK1, RANKL and SOST. *Conclusions*: The STING-P-TBK1 signaling pathway plays a crucial role in the regulation of the secretion of inflammatory cytokines and osteoclastogenesis potential in IR-induced bone microenvironment disruption. The selective STING antagonist can be used to intervene to block the STING pathway and, thereby, repair IR-induced multicellular biological damage and mitigate the imbalance between osteoclastogenesis and osteoblastgenesis.

## 1. Introduction

Clinical radiotherapy is one of the most effective and indispensable methods for cancer treatment modalities, but ionizing radiation inevitably causes damage to normal tissues near the tumor, including bone tissues [1,2,3]. Indeed, demineralization of bone, thinning of bones, sclerosis and loss of trabecular connections have been described following radiotherapy [4,5,6]. Patients with cervical, endometrial and prostate cancer may develop pelvic insufficiency fractures post-radiotherapy, which are difficult to treat and are associated with very high rates of delayed union and non-union [7,8]. It has been reported that the incidence of rib fracture in breast cancer patients one year after radiotherapy can reach 1.8~19% [9]. Postmenopausal women who receive radiotherapy for cervical cancer, rectal cancer and proximal anal cancer have an increased risk of developing pelvic fractures compared to non-radiotherapy patients, with risk ratios of 1.66, 1.65 and 3.16, respectively [10]. The incidence of hip fractures in prostate cancer patients after radiotherapy increased by 76% [8]. In clinical practice, a spectrum of bone changes from mild osteopenia to osteoradionecrosis is often observed post-radiotherapy, which could seriously affect the cancer survivor’s quality of life and the healthcare cost [11,12]. Meanwhile, certain studies have demonstrated that radiotherapy can not only cause bone damage in the irradiated area but also in the remote bone tissues that are not directly irradiated, which is referred to as the distal effect and systemic response of irradiation (IR)-induced bone injury [13,14]. Moreover, IR-induced bone microenvironment disruption, which is mainly manifested as the activation of osteoclastogenesis and restraint of osteogenesis, creates a more favorable environment for tumor metastases [15,16,17]. To date, the mechanism of IR-induced bone disruption has not been fully elucidated. Increasing evidence suggests that cellular senescence is involved in the pathogenesis and development of IR-induced skeletal fragility, while the bone marrow (BM) microenvironment has a crucial regulatory effect in the process of these diseases [18,19]. In order to effectively prevent radiotherapy-associated side effects in bone tissue and its microenvironment, it is particularly important to explore the pathophysiology of IR-induced bone disruption and the cellular regulatory mechanism for an imbalance in bone homeostasis.

Damage to osteoblasts and osteocytes (OCYs) within the bone microenvironment is believed to be a primary contributor to decreasing bone mineral density following irradiation [6,20,21]. Evidence has shown that in response to cellular stressors, including radiotherapy, chemotherapy and metabolic dysfunction, multiple cell types in the bone microenvironment undergo functional alterations and become senescent, with osteocytes and myeloid cells as primary contributors to the heterogeneous SASP [18]. Ionizing radiation can induce oxidative stress and DNA damage, which are physiological inducers of cellular senescence [22,23,24,25]. Senescent cells can accumulate in excess chronically at the sites of tissues, as well as having local and likely systemic effects through their release of a pro-inflammatory secretome known as the senescence-associated secretory phenotype (SASP) [26,27,28]. Senescent osteocytes are potential therapeutic targets to alleviate skeletal dysfunction. It has been demonstrated that cellular senescence and its SASP contribute to radiotherapy-induced bone loss that can be rescued by depleting senescent cells or blocking the deleterious secretome [26]. Previous studies consistently demonstrate that SASPs are regulated by multiple pathways, such as the MAPK/NF-κB signaling pathway, the cGAS-STING signaling pathway and the JAK/STAT signaling pathway. As a member of the cGAS-STING signaling pathway, the stimulator of interferon genes (STING) is an intracellular signaling molecule located in the outer membrane of the endoplasmic reticulum, mitochondria and microsomes of host cells [29]. Signal sensors closely related to the innate immune response are essential for the production of type I interferons (IFNs) and subsequent inflammatory immune responses [30,31].

Meanwhile, the expression of a variety of inflammatory cytokines is regulated by the STING pathway, such as TNF-α and IL-6, by activating the NF-κB signaling pathway [30,32,33], which is highly correlated with bone metabolism regulation, including the recruitment and differentiation of osteoclast precursors. The metabolism imbalance caused by oxidative stress in the bone microenvironment is affected by a variety of inflammatory secretory phenotypes, including IL-1α, IL-6 and NF-κB. Therefore, the activation of the STING pathway of bone microenvironment cells, especially OCYs, which are prone to accumulating damage and lead to premature senescence as long-lived cells, may play a key regulatory function in the stress-associated imbalance of bone homeostasis [33]. Radiotherapy-induced SASP secretion can be limited by targeting the STING pathway, resulting in preservation of bone integrity in radiotherapy-treated mice [30,31,34]. In recent years, with a further understanding of the role of STING in activating immune surveillance, a variety of small-molecule inhibitors, such as C-176, have emerged, which can significantly decrease the STING-mediated downstream IFN-β activity and effectively ameliorate systemic inflammation in mice [32,33,35]. C-176 is a strong and covalent mouse STING inhibitor. The good efficacy and no obvious toxicological influences have laid the foundation for potential therapeutic interventions in the future [36,37,38]. In the present study, both a mouse model of bone degenerative injury via single local IR of the hind limb and an in vitro IR-induced senescent OCY model were constructed. Targeted interventions blocking the STING pathway were conducted in order to elucidate whether ionizing radiation could activate the STING-phosphorylated TBK1 (P-TBK1) pathway and enhance SASP secretion by senescent OCYs, thereby promoting osteoclastogenesis through paracrine signaling of inflammatory factors as an intercellular messenger. In this study, we hypothesize the physiological as well as pathological processes during IR–induced bone homeostasis deterioration in order to establish a broad understanding of the biomechanical as well as molecular mechanisms involved in senescent microenvironment and bone homeostasis deterioration. In addition, evidence for the use of STING inhibitors as potential therapeutic agents targeting bone resorption in IR-induced bone microenvironment disruption will be discussed. We highlight the diversity of the senescent cells in the microenvironment of bone, as well as the mechanisms by which these senescent cells are involved in musculoskeletal diseases, such as IR-induced bone homeostasis deterioration.

## 2. Materials and Methods

### 2.1. Animals and Cell Lines

Male 6-week-old BALB/c mice (weight, ~22 g) were purchased from JieSijie Laboratory Animal Co., Ltd.(Shanghai, China). All mice were housed in specific pathogen-free and temperature-, moisture- and light-controlled (12 h light/dark cycle) cages, and they were fed with standard laboratory chow with access to water ad libitum. All mouse experiments complied with relevant ethical regulations and were approved by The Experimental Animal Ethics Committee of Fudan University (Shanghai, China).

The mouse osteocyte-like MLO-Y4 cells, a murine-derived model of an OCY [39], and RAW264.7, a mouse macrophage cell line, were obtained from Cell Bank (Shanghai Institutes for Biological Sciences, Chinese Academy of Sciences, Shanghai, China). The cell lines were cultured on collagen type-I-coated plates (Solarbio Science & Technology Co., Ltd., Beijing, China) in α-MEM (Gibco, Eggenstein, Germany) supplemented with 10% fetal bovine serum (FBS; Gibco, Eggenstein, Germany) and 1% penicillin/streptomycin (PS; Sigma-Aldrich, St. Louis, MO, USA) at 37 °C in 5% CO_2_. The cell lines were sub-cultured approximately every 72 h to maintain the cell populations [40]. 

### 2.2. Reagents

The following reagents were purchased: lipofectamine RNAiMAX (cat. no. 13778100; Thermo Fisher Scientific, Inc., Waltham, MA, USA); opti-MEM (cat. no. 31985062; Gibco, Eggenstein, Germany); STING small interfering RNA (siRNA), negative control (NC)-siRNA (GenePharma Co., Ltd., Shanghai, China); STING small-molecule inhibitor, C-176 (cat. no. HY-112906; Med-ChemExpress, Princeton, NJ, USA); acid phosphatase, leukocyte (tartrate-resistant acid phosphatase; TRAP) Kit (cat. no. 387A-1KT; Sigma-Aldrich, St. Louis, MO, USA); mouse cytokine array (cat. no. ARY028; R&D Systems, Inc., Minneapolis, MN, USA).

The following reagents were purchased from Beyotime Institute of Biotechnology: Western Primary Antibody Dilution Buffer (cat. no. P0023A), Western Secondary Antibody Dilution Buffer (cat. no. P0023D), BeyoECL Star Super Sensitive ECL Chemiluminescence Kit (cat. no. P0018AS), BCA Protein Assay Kit (Cat. no. P0012) and Senescence β-Galactosidase Staining Kit (cat. no. C0602).

### 2.3. Mouse Model of Local IR and C-176 Intervention

Firstly, 24 6-week-old BALB/c mice were adaptively reared for 1 week and randomly divided into two groups, the control group and the local IR group, with 12 mice in each group. On day 0, all mice were anesthetized via intraperitoneal injection of 1.0% sodium pentobarbital (30 mg/kg body weight) and fixed in a supine position on a self-designed wooden board. This board was subsequently placed in the IR cavity, and the left hind limb of each mouse was exposed in the center of the circular hole. The local IR mouse group was tested using an X-Rad 320 Biological Irradiator (X-RAD 320, PXi) with a collimator (diameter, 1.8 cm). The dose rate was 121.3 cGy/min, with 220 kV tube voltage, 12 mA tube current and a distance of 60 cm between the source and the surface. A single dose of 12 Gy was delivered to a 5 mm region of the distal metaphyseal region of the left hind limb (termed the ‘local IR limb’), while the right hind limb served as the contralateral control (termed the ‘contralateral limb’). The non-irradiated hind limbs of the control group mice were termed ‘control limb’. On day 8, experimental mice were injected intraperitoneally with 1% sodium pentobarbital (50 mg/kg body weight). After collecting 0.5 mL peripheral blood sample from the orbit under anesthesia, the mice were sacrificed via cervical dislocation, and samples, including bone tissue and bone marrow, were collected and analyzed.

Next, another 36 male BALB/c mice were administered the STING small-molecule inhibitor C-176 for intervention experiment. All mice were randomly divided into three groups: the control group, the local IR group and the local IR + C-176 intervention group, with 12 mice in each group. The mice in the local IR group were irradiated with 12 Gy on day 0, while the mice in the local IR + C-176 intervention group were administered with intraperitoneal injections of C-176 (750 nmol C-176 per mouse in 200 μL corn oil; 13 mg/kg body weight) 1–7 days after irradiation, and the other two groups were administered 200 μL corn oil. On day 8, the experimental mice were sacrificed, and samples of blood and bone were collected as described above.

### 2.4. Decalcified Bone Tissue Embedding and Histology

Thus, 7 days after IR, the femurs of the mice were harvested, fixed with 4% paraformaldehyde for 24 h, decalcified in EDTA, embedded in paraffin and cut in the vertical plane at 4 μm. The sections were stained in hematoxylin–eosin (HE) for 3 min at room temperature, the volume of bone marrow adipocytes (BMAs) was measured to evaluate the adipogenesis changes, and the number of OCYs lacunae in cortical bone was measured to evaluate the IR-induced OCYs exfoliation.

Next, TRAP staining of bone histological sections was conducted under the instruction of the TRAP staining kit, and the TRAP^+^ area was visualized and calculated to evaluate the osteoclastogenesis potential. Briefly, the bone tissue section was incubated with prepared TRAP staining solution for 2 h at 37 °C in the dark, then washed with distilled water and treated with hematoxylin staining solution for 5 min at room temperature to counterstain the nuclei, washed with distilled water, and then imaged using an optical microscope (Leica DMI3000B; ×100 magnification).

In addition, the femoral paraffin sections underwent immunohistochemistry staining of senescence-related proteins, p21 and p16, in order to assess for senescence-related alterations in the bone microenvironment. Briefly, the tissue sections were deparaffinized and hydrated and then heated in citrate antigen retrieval solution (cat.no. P0081; Beyotime Institute of Biotechnology) at 100 °C for 20 min, and the tissues were incubated and blocked with 5% BSA (cat.no. V900933; Sigma-Aldrich; Merck KGaA) for 30 min. Primary p16 antibody (cat. no. ab51243; Abcam, Cambridge, UK; 1:1000) and p21 antibody (cat. no. ab109199; Abcam; 1:1000) were then incubated with the sections overnight at 4 °C. Next, sections were washed with TBST (TBS supplemented with 0.1% Tween 20) three times and incubated with anti-rabbit secondary antibody (cat. no. A0208; Proteintech Group, Inc., Rosemont, PA, USA; 1:1000) for 1 h at room temperature. After washing with TBST, sections were added to DAB staining solution and incubated at room temperature, while nuclei were counterstained with hematoxylin for 5 min at room temperature. Images of the sections were taken under a microscope, and the proportion of OCYs that were positive for the p16 and p21 marker proteins in the bone cortex was calculated.

### 2.5. Cytokine Antibody Microarray Analysis to Identify SASP Composition

The mouse cytokine antibody chip array detection kit (Mouse Cytokine Array; cat. no. ARY028; R&D Systems, Inc., Minneapolis, MN, USA) was used to detect the changes in protein levels of 111 cytokines and chemokines in the serum of locally irradiated mice. Briefly, 200 μL mouse serum was diluted with the corresponding array buffer. The diluted mouse serum was added to the protein-blocked antibody array membrane and incubated overnight at 4 °C. Next, the membrane was washed three times with the corresponding washing buffer solution, and streptavidin-HRP was incubated with the membrane for 30 min and shaken at room temperature. The membrane was then washed three times with the washing buffer solution, and a chromogenic reagent was added. The signal was detected using a luminescence imager (Chemi Scope 6300, Qinxiang Scientific Instrument Co., Shanghai, China), and the pixel density was quantified with HLImage++ 6.2 computer vision systems (1997, Western Vision Software, Salt Lake City, UT, USA). When the value of corrected pixel density was 2 standard deviations higher or lower than the control group, the index was defined as up- or downregulation. 

### 2.6. Isolation of Primary OCYs for Culture and Subsequent Analysis

A modified sequential digestion method was used to isolate primary OCYs from the femur and tibia of BALB/c mice for in vitro culture and subsequent experiments. Briefly, femora and tibiae were aseptically dissected and immersed in 75% ethanol, placed in sterile PBS containing 10% penicillin and streptomycin, and then processed via serial digestion using 0.001 g/mL collagenase type Ι (Sigma-Aldrich, St. Louis, MO, USA), 0.001 g/mL collagenase type II (Sigma-Aldrich, St. Louis, MO, USA) and 0.25% trypsin-EDTA dissolved in α-MEM supplemented with 10% FBS and 1% PS [15,16,41,42,43]. The identification of primary OCYs was based on cell morphology with slender synapses, E11-specific protein expression and weak ALP expression [15,16].

The bone fragments were incubated with EDTA solution for 30 min; then, the solution was aspirated for culturing OCY cells. The aforementioned procedures were repeated 2 times, and the combined cell suspension solution was centrifuged at 200× *g* for 10 min at room temperature, followed by the removal of supernatant from the cell pellet and resuspension of the cells in the culture medium. The remaining bone fragments were evenly plated into 6 cm diameter dishes before the cell suspension (after re-suspension several times) was added evenly. Culture medium was added to the cells, which were incubated at 37 °C in 5% CO_2_, for subsequent proliferation.

Cell viability of the primary OCYs was detected using Cell Counting Kit-8 (CCK-8; cat. no. CK04; Dojindo, Kumamoto, Japan). Cells were incubated with 10 μL CCK-8 solution and 100 μL α-MEM for ~2 h at 37 °C under dark conditions. Cell viability was calculated by measuring absorbance at 450 nm using an enzyme standardizer (BioTek ELX800). The experiment was repeated three times, with six replicate wells for each group.

Next, senescence-associated β-galactosidase (SA-β-gal) staining of mouse primary OCYs was performed using an SA-β-gal staining kit (cat. no. C0602; Beyotime Institute of Biotechnology, Shanghai, China). Firstly, the culture supernatant was discarded, the cells were washed once with PBS and 4% paraformaldehyde was added to fix the cells at room temperature for 15 min. Next, the mixed and prepared staining working solution was incubated with the cells for 24 h at 37 °C, without CO_2_. Finally, the staining solution was removed, and cells were washed once with PBS and observed under an optical microscope. The percentage of SA-β-gal-positive cells was calculated using Image J software (Version 1.53; National Institutes of Health, Bethesda, MD, USA).

### 2.7. Isolation of Mouse Bone Marrow Mononuclear Macrophages (BMMs) for Culture and Subsequent Analysis

To evaluate the changes in osteoclastogenesis potential via IR, BMMs were harvested from experimentally irradiated and non-irradiated mice, 7 days post-irradiation. Ficoll-Paque PREMIUM density medium (cat. no. 17544602; Cytiva, Inc., Marlborough, MA, USA) and an equal volume of bone marrow suspension (collected by flushing long bone marrow cavity with α-MEM medium) were added slowly to a centrifuge tube that was centrifuged at 1200× *g* for 30 min. After centrifugation, the cells in the albuginea layer between the upper and lower layers were aspirated gently with a pipette and centrifuged twice at 200× *g* for 10 min in PBS buffer. After removal of the supernatant, the cells were resuspended in α-MEM medium and transferred to a 25T culture flask to continue cell culturing. Non-adherent cells were collected after 24 h and resuspended in α-MEM supplemented with 25 ng/mL macrophage colony-stimulating factor (M-CSF; cat. no. 315-20; PeproTech, Inc., Rocky Hill, NJ, USA), 50 ng/mL RANKL (cat. no. 315-11; PeproTech, Inc., Rocky Hill, NJ, USA), 10% FBS and 1% PS, and they were inoculated in well plates. The osteoclast inducible medium was changed every 3 days. The fusion of multinucleated macrophages, which were mature osteoclasts, was visualized under an optical microscope (Leica DMI3000B).

The ability of osteoclast formation was evaluated in terms of the number and area of TRAP^+^ cells. Briefly, the cells were fixed with 2.5% glutaraldehyde for 10 min at room temperature and stained under the instruction of the TRAP staining kit. The TRAP staining protocol was the same as that in the aforementioned Decalcified bone tissue embedding and histology sub-section.

### 2.8. Isolation of Mouse Bone Marrow Mesenchymal Stem Cells (BMSCs) for Culture and Subsequent Analysis

The bone marrow of the mice’s femur and tibia was flushed out with α-MEM using 1 mL syringes and was harvested to culture BMSCs. After 48 h, non-adherent cells were discarded, and the adherent cells were defined as the first-generation cells (passage 1), when the cells grew to 80% confluency [44]. Cells of passage 3 were used for all experiments.

The osteogenic ability of BMSCs was evaluated via alkaline phosphatase (ALP) staining. The BMSCs isolated from the irradiated and non-irradiated mice were cultured in vitro. After 24 h, the culture medium was replaced with osteogenic induction medium (α-MEM comprising 50 mg/L ascorbic acid, 0.01 μM dexamethasone, 10 mM β-glycerophosphate, 15% FBS and 1% PS), followed by continuous culturing for 7 days. Subsequently, ALP was determined using a BCIP/NBT Alkaline Phosphatase Color Development Kit (cat. no. C3206; Beyotime Institute of Biotechnology, Shanghai, China). Briefly, the supernatant was discarded, and the cells were washed twice with PBS. Then, 2.5% glutaraldehyde was added to fix the cells for 5 min at temperature, and the prepared BCIP/NBT alkaline phosphatase staining solution was incubated with the cells for 1 h at 37 °C in the dark. Cells were washed once with PBS and imaged under an optical microscope (Leica DMI3000B).

### 2.9. In Vitro OCY Model of IR

OCY-like cells (MLO-Y4) were irradiated with a total dose of 2, 4 or 8 Gy using ^137^Cs γ-rays (Nordion, Ottawa, ON, Canada) at a dose rate of 66.7 cGy/min. For the control group, sham-irradiated cells were placed in the irradiator for the same duration but received no radiation. After the IR, the medium was changed every other day, and the cells were cultured and collected for follow-up experiments.

### 2.10. siRNA Transfection

MLO-Y4 cells were inoculated with α-MEM medium at 50% confluency and exposed to ^137^Cs γ-rays (Nordion, Ottawa, ON, Canada). The culture medium was changed to a complete medium without antibiotics. After stabilizing in an incubator at 37 °C for 30–60 min, cells were transfected with siRNA (the sequences of the STING siRNA: 5′-UCAAUCAGCUACAUAACAA-3′; the sequences of the negative control siRNA: 5′-UUCUCCGAACGUGUCACGUTT-3′) and transfection reagent Lipofectamine RNAiMAX (25 pmol per well) for 6 h at 37 °C. The base medium was changed to fresh medium containing antibiotics. After 48 h, the cells and culture supernatant were collected and temporarily stored at −80 °C.

### 2.11. Co-Culture of Cell-Conditioned Medium (CM) and RAW264.7 Cells

The MLO-Y4 cells cultured in vitro after IR and siRNA transfection were cultured for 1 day. The culture medium was substituted with complete media without antibiotics and FBS, and cells were cultured at 37 °C for 1 day. Subsequently, the cell media supernatant was collected and filtered with a 0.22 μm suction filter and termed ‘CM’. This CM was sub-packaged in centrifuge tubes and stored at −80 °C. Subsequently, RAW264.7 cells were incubated in plates with α-MEM medium containing 25 ng/mL RANKL for 24 h. Next, the culture medium was replaced with α-MEM medium supplemented with 25 ng/mL RANKL and 25% CM every 2 days. On the 7th day, the cells were stained using a TRAP staining kit and visualized under the optical microscope (Leica DMI 3000B). The TRAP^+^ multinucleated mature osteoclasts were observed and calculated using Image J software (Version 1.53; National Institutes of Health, Bethesda, MD, USA).

### 2.12. Western Blot Analysis

Cell disintegration was extracted using RIPA lysis buffer (cat. no. P0013B; Beyotime Institute of Biotechnology, Shanghai, China) supplemented with the complete protease inhibitor, phenylmethanesulfonyl fluoride (Beyotime Institute of Biotechnology, Shanghai, China). The quantification of total protein concentration was determined using the BCA Protein Assay Kit according to the manufacturer’s instructions. Proteins were diluted in NuPAGE loading dye, heated at 95 °C for 7 min and separated using 12.5% SDS-PAGE (cat. no. PG113; Epizyme Biotech Co., Ltd., Shanghai, China). Following the electrophoretic transfer of proteins onto PVDF membranes (Millipore, Billerica, MA, USA), non-specific binding was blocked by incubation with 5% skimmed milk for 30 min at room temperature. Next, membranes were incubated with the following primary antibodies: anti-STING (cat. no. 13647; Cell Signaling Technology, Inc., Danvers, MA, USA; 1:1000) anti-P-TBK1 (cat. no. 5483; Cell Signaling Technology, Inc., Danvers, MA, USA; 1:1000), anti-RANKL (cat. no. ab45039; Abcam; 1:1000), anti-osteoprotegerin (OPG; cat. no. ab183910; Abcam; 1:1000), anti-sclerostin (SOST; cat. no. ab63097; Abcam; 1:1000) or anti-β-actin (cat. no. AF7018; Cell Signaling Technology, Inc., Danvers, MA, USA; 1:1000). Membranes were then washed with TBST and incubated with the appropriate HRP-conjugated secondary antibody (anti-rabbit (cat. no. A0208) or anti-mouse (cat. no. A0216; Proteintech Group, Inc., Rosemont, PA, USA; 1:2000)) for 1 h at room temperature. Proteins were visualized using an ECL Kit and an Omega Lum™C imaging system. Quantitative analysis was conducted using Image J software (Version 1.53; National Institutes of Health, Bethesda, MD, USA).

### 2.13. RNA Extraction and Reverse-Transcription–Quantitative PCR (RT-qPCR)

For the analysis of mRNA expression levels, total RNA was isolated using the Simply P Total RNA Extraction Kit (cat. no. BSC52S1; Bioer Technology, Beijing, China) according to the manufacturer’s protocol. The isolated RNA was reverse transcribed at 42 °C for 15 min and 95 °C for 3 min into cDNA using FastKing gDNA Dispelling RT SuperMix (cat. no. KR118-02; Tiangen Biotech Co., Ltd., Beijing, China). qPCR was performed on the ABI QuantStudio 5 (Applied Biosystems, Carlsbad, CA, USA) with PowerUp SYBR Green Master Mix (Invitrogen; Thermo Fisher Scientific, Inc., Waltham, MA, USA) in 10 μL PCR reaction system (Table 1). The thermocycling conditions used were: 40 cycles of 95 °C for 15 s followed by 55 °C for 15 s and 72 °C for 1 min. The 2^−ΔΔCq^ method was used to quantify the relative mRNA expression levels of the indicated genes [45]. The primer sequences are presented in Table 2.

### 2.14. Statistical Analysis

Statistical significance was performed using SPSS Statistics 16.0 software (SPSS, Inc., Chicago, IL, USA), and GraphPad Prism 8.2.1 software (Dotmatics) was used for the generation of all curves and statistical graphs. Data are presented as mean ± standard deviation. The n values represent the number of independent experiments performed or the number of individual mice per group. One-way ANOVA analysis with the Tukey–Kramer post hoc test was used for multiple group comparisons. Statistical significance of C-176 interventional data was determined using two-way ANOVA, followed by Bonferroni’s post hoc test. *p* < 0.05 was considered to indicate a statistically significant difference.

## 3. Results

### 3.1. Radiation Induces Multicellular Dysfunction in the Bone Microenvironment

Bone is a dynamic tissue; a variety of bone cells, such as BMSCs, osteoblasts, OCYs, hematopoietic stem cells and osteoclasts, are involved in the process of bone remodeling. Bone homeostasis, which is maintained by an intricate balance between bone formation by osteoblasts and bone resorption by osteoclasts, plays an important role in maintaining bone integrity and strength. In order to investigate the IR-induced biological injury of bone cells in mice, the present study was designed to establish a mouse model of local IR. The biological alterations in bone marrow cells from locally irradiated and contralateral hind limbs were analyzed at 7 days post-radiation. Following single 12 Gy local irradiation, all experimental mice survived, and during the study period, the mice weight in the IR group exhibited no significant change compared with the control group. In addition, lumbar aBMD decreased at 7 days after single 12 Gy local irradiation, but the differences were not statistically significant compared with the control groups (detailed data not shown). As mentioned, no significant overall adverse effects were observed on the irradiated mice during the experimental period. The osteogenic and adipogenic differentiation potential of BMSCs has a key influence on the behavior of bone formation. In the present study, to assess the adipogenesis changes in the bone marrow, HE staining of femoral tissue specimens of mice was used to evaluate the volume of BMAs. The results demonstrated that, compared with mice in the control group (without IR), the volume of BMAs of the locally irradiated hind limb was significantly increased (*p* < 0.001), even occupying as much as 30% of the bone marrow cavity (Figure 1(A1)). The volume of BMAs of the contralateral hind limb that was not directly exposed also increased markedly compared with that of the control mice, but the increase was a less significant increment (Figure 1(A1)). As mentioned above, the adverse effects of IR on bone marrow are mainly on red bone marrow, which was enriched with hematopoietic tissue, rather than yellow bone marrow, which was enriched with adipose tissue, manifested as degeneration of osteogenesis and fate conversion of BMSCs into adipocytes.

In addition, to explore the effect of 12 Gy local IR on the osteogenic differentiation potential, the BMSCs of the locally irradiated and contralateral hind limbs were harvested and cultured for osteogenic differentiation in vitro. The ALP staining demonstrated an obvious decrease in APL expression in the locally irradiated hind limb compared with that of the control group (Figure 1(B1)). The same trend was observed in the BMSCs of the contralateral hind limbs (*p* < 0.01; Figure 1(B1)), suggesting that 12 Gy local IR may lead to a significant decrease in the osteogenic differentiation potential of BMSCs in the locally irradiated and the not directly exposed areas.

Osteoclasts play a key role in bone resorption and can express TRAP. The positive expression of TRAP in bone tissue is an important histologic marker of osteoclast activity and bone resorption. Compared with the control mice, the cancellous bone area of the locally irradiated hind limb demonstrated a considerable increase in TRAP expression (*p* < 0.001), and the contralateral hind limb also demonstrated an increase in osteoclast activity (*p* < 0.05; Figure 1(A2)), suggesting a transient enhancement in OCs that mobilize into bone tissue through the stress response of IR.

In addition, BMMs from the hind limb of the mice were harvested at 7 days post-radiation exposure, and subsequent osteoclast differentiation was induced in vitro. It was demonstrated that the osteoclastogenic capacity of BMMs in the contralateral hind limb of the irradiated mice group was significantly promoted compared with that of the control mice, as reflected by the increased number of TRAP^+^ osteoclasts and the larger positive fusion area (*p* < 0.01; Figure 1(B2)). However, the number of mature osteoclasts derived from the bone marrow of the locally irradiated hind limb was low, which may be due to the damage of radiosensitive bone marrow after direct IR exposure, resulting in a decrease in the myeloid progenitors (Figure 1(B2)).

As the main coordinator of bone remodeling, OCYs are the most abundant cells in bone tissue and participate in the formation of bone tissue. In the present study, HE staining of bone tissue sections revealed that, compared with the control group, the number of cell lacunae in bone tissues was significantly higher, both in the locally irradiated and in the contralateral hind limbs (*p* < 0.05; Figure 1(A3)), which indicates an increase in OCYs exfoliation. It is known that the number and viability of OCYs are important to ensure bone structure integrity and the function of the bone remodeling network. In the present study, the long bone tissues of mice were collected, and the primary OCYs were obtained using the modified sequential digestion method established in our laboratory’s previous research [15,16]. The identification of primary OCYs was based on cell morphology with slender synapses under an optical and electron microscope, E11-specific protein expression through Western blot analyses and weak ALP expression via ALP staining. The cell purity reaches over 95% and can be used for in vitro culture and subsequent experiments. Compared with the control mice, the number of OCYs derived from irradiated mice was significantly decreased, accompanied by decreased cell viability (Figure 1(B3)). Meanwhile, the cell morphology also exhibited obvious changes, with a shortened dendritic branch structure, cytoplasmic disorder and sparseness, particularly in OCYs derived from the locally irradiated hind limb (Figure 1(B3)).

Collectively, the results suggest that 12 Gy local IR does induce multi-cellular dysfunction of the bone microenvironment, both in the locally irradiated and contralateral area and the non-irradiated area. This is manifested as an enhancement in OCYs exfoliation, activation of osteoclastogenesis, degeneration of osteogenesis and fate conversion of BMSCs into adipocytes.

### 3.2. Radiation Induces Bone Aging and Enhances SASP Secretion

Radiation is an inducer of DNA damage that can, in turn, induce cellular senescence. To assess the senescence-related alterations in cell populations in the bone microenvironment, the expression of the senescence markers, p16^INK4a^ (p16) and p21^WAF1/CIP1^ (p21), was analyzed via immunohistochemistry of femoral sections 7 days post-radiation. The results demonstrated that the positive expression of p16 and p21 was increased in the cortical bone matrix of irradiated mice compared with the control group (Figure 2A). The enhancement in positive p16 and p21 expression was more notable in the tissues of the locally irradiated hind limb, which demonstrated that ionizing radiation may induce the senescent phenotype of bone tissue cells (*p* < 0.01; Figure 2A).

To further verify the senescence phenotype of OCYs in locally irradiated mice, primary OCYs isolated and cultured from the hind limb of irradiated and non-irradiated mice were stained with SA-β-gal, a well-established marker of senescence. Compared with the control group, the proportion of SA-β-gal^+^ OCYs derived from the locally irradiated and the contralateral hind limbs of the irradiated mice was significantly increased (Figure 2B).

Having found evidence of senescence in the bone tissue and cells following IR, it was next elucidated whether SASP, a typical feature of cell aging, has an important influence on the IR-induced damage of cell biological function. A semi-quantitative microarray was performed on the serum of experimental mice to analyze the differential expression changes in typical components of SASP. The results demonstrated that the expression of serum SASP factors was significantly altered when mice received 12 Gy local IR, compared with the control group (Figure 3A). Among the 111 cytokines detected, the expression of 53 cytokines was significantly upregulated in the serum of the irradiated mice. A wide array of SASP factors, which play a crucial role in the regulation of bone homeostasis, mainly including the interleukin family (IL-7, IL-33, IL-12p40), chemokines (such as CXCL16, CCL11, CCL21), inflammatory factors (IGFBP6, Resistin, DPPIV) and metalloproteinases (MMP-9, MMP-2, MMP-3), all showed some degree of elevated expression, as detailed in Table 3. The changes in the expression of the most variable cytokines are shown in a heatmap in Figure 3B.

Collectively, these results provide evidence that radiation-induced bone aging is accompanied by altered SASP secretion and the upregulated expression of multiple cytokines.

### 3.3. Radiation Activates the STING-P-TBK1 Pathway and Promotes Osteoclastogenesis via Paracrine Signaling

Given the aforementioned findings indicating that radiotherapy is sufficient to induce senescence, it was next elucidated whether the STING-P-TBK1 pathway was activated in the bone tissue of irradiated mice.

In the present study, the potential role of STING in the progression of inflammation factor production and osteoclastogenesis was investigated using a 12 Gy locally irradiated mice model and an OCY MLO-Y4 in vitro model. It was found that the expression of STING in primary OCYs harvested from the locally irradiated hind limbs of mice was significantly elevated, and a similar upregulation of STING expression was observed in the OCYs isolated from contralateral hind limbs (Figure 4A). Accordingly, the levels of P-TBK1, which is the direct downstream signaling protein of STING, were also significantly increased in locally irradiated mice, which indicated that the STING pathway was activated in the bone tissue of locally irradiated mice (Figure 4A). 

Consistent with the activation of the STING-P-TBK1 pathway, the expression levels of the relevant functional proteins secreted by OCYs were also altered (Figure 4B). It was found that, compared with the mice in the control group, the expression levels of RANKL and SOST in OCYs derived from the locally irradiated and contralateral hind limbs were significantly increased (*p* < 0.05 and *p* < 0.001, respectively). Meanwhile, the protein expression level of OPG in OCYs derived from the locally irradiated hind limb was significantly decreased (*p* < 0.01). The ratio of RANKL / OPG expression levels in OCYs from the locally irradiated and contralateral hind limbs was significantly higher than in the control group, indicating activation of bone resorption (Figure 4B; *p* < 0.05 and *p* < 0.01, respectively). It is, therefore, suggested that ionizing radiation could alter the expression of related functional proteins secreted by OCYs by activating the STING-P-TBK1 pathway (Figure 4B). Next, in vitro experiments were conducted to verify the activation of the STING pathway in OCY-like cells (MLO-Y4) after IR. Similar results were obtained when MLO-Y4 cells were exposed to 2, 4 and 8 Gy radiation (Figure 4C). Western blot results demonstrated that the expression of STING was enhanced significantly following IR, accompanied by the upregulated expression of P-TBK1 downstream of the pathway, especially in the 4 Gy radiation group (Figure 4C). In subsequent experiments, an intermediate dose of 4 Gy was adopted to induce senescence in MLO-Y4 cells and inflammatory secretion.

Furthermore, an in vitro transfection experiment in MLO-Y4 cells was performed to assess whether the observed susceptivity to IR-induced inflammatory factor production might be associated with the altered expression of proteins involved in the pathways downstream of STING and P-TBK1 stimulation. In vitro cultured MLO-Y4 cells were exposed to 4 Gy IR and subsequently transfected with STING-siRNA 3 days post-IR, to knockdown the expression of STING. The results demonstrated that a beneficial recovery with IR-induced upregulation of P-TBK1 protein expression was observed after STING-siRNA knocked down the STING (Figure 5B). It was subsequently investigated whether RANKL was affected by the STING-P-TBK1 pathway after IR, and the results demonstrated that the protein expression of RANKL in irradiated MLO-Y4 cells was significantly enhanced (Figure 5B). However, a corresponding decrease in IR-induced activation of RANKL was accompanied by a downregulation in P-TBK1 protein expression following STING-siRNA intervention, confirming the crucial role of STING in RANKL secretion (Figure 5B). Similarly, the mRNA expression levels of inflammatory factors, such as IL-1α, IL-6 and NF-κB, which are involved in the modulation of the osteoclastogenesis, were significantly upregulated in irradiated OCYs (Figure 5C). However, a recovery of downregulation of IR-induced activation of the IL-1α, IL-6 and NF-kB genes was detected after STING-siRNA knocked down the STING pathway (*p* < 0.05; Figure 5C). These results suggest that the STING-P-TBK1 signaling pathway plays a crucial role in the regulation of inflammatory factor secretion in IR-induced bone microenvironment disruption.

It was hypothesized that it is this permissive effect of an activated STING pathway on the inflammatory cascade that is responsible for the activation of osteoclasts and the IR-induced bone microenvironment disruption. Subsequently, the paracrine regulation effect on osteoclastogenesis by the IR-induced inflammatory cytokine of OCYs was investigated. The CM of MLO-Y4 cells was collected 3 days after 0 Gy or 4 Gy IR, with or without STING-siRNA intervention. This CM was co-cultured with RAW264.7 cells and then TRAP^+^ cells were observed. The results suggested a significant promotion in the osteoclast differentiation ability of RAW264.7 cells co-cultured with the CM of irradiated MLO-Y4 cells (CM-4Gy-NC), compared with the control group (CM-0Gy-NC), which was indicated by a significant increase in the number of TRAP^+^ cells (*p* < 0.001, Figure 5D). When the STING pathway was knocked down with STING-siRNA-targeted intervention, the excessive activation of osteoclastogenesis by the CM of irradiated MLO-Y4 cells was significantly reversed, which was indicated by a significant decrease in the number of TRAP^+^ cells in the CM-4Gy-SI group, which is RAW 264.7 cells cocultured with the CM of irradiated STING-targeted transfection of MLO-Y4 cells (*p* < 0.01; Figure 5D).

Collectively, these results suggest that the STING gene may be a logical candidate to explain the excessive activation of inflammatory cytokines, including RANKL and IL-1α, in addition to directly activating osteoclastogenesis in response to IR-induced bone microenvironment disruption.

### 3.4. STING Inhibitor Ameliorates IR-Induced Multicellular Biological Disruption

It was next elucidated whether targeting STING could protect against IR-induced bone microenvironment disruption and the activation of osteoclastogenesis. To address this, a STING inhibitor, C-176, was administered to mice pre-treated with 12 Gy local IR. In the present study, mice were randomly divided into three groups with 12 mice per group: control group, local IR group and local IR + C-176 intervention group (Figure 6A). 

During the experiment, the body weight of the three mice groups all increased, without notable differences among the three groups (Figure 6B), indicating that C-176 had no toxic effect on the mice. At the endpoint of the C-176 intervention experiment, all experimental mice survived, and there was some recovery of lumbar aBMD after intervention with C-176 compared to the local IR group, but the difference was not significant (detailed data not shown). On the eighth day of IR and C-176 administration, primary OCYs were extracted from the hind limbs of the mice (with or without 12 Gy single local IR) and cultured in vitro for experimentation. For the expressions of the pathway-related proteins STING and P-TBK1, two-way ANOVA demonstrated significant effects of irradiation (F = 210.921, *p* < 0.001; F = 48.570, *p* < 0.001) and C-176 intervention (F = 168.144, *p* < 0.001; F = 28.746, *p* < 0.001). Meanwhile, for the bone resorption-related proteins RANKL and SOST, two-way ANOVA demonstrated significant effects of irradiation (F = 45.013, *p* < 0.001; F = 153.481, *p* < 0.001) and significant effects of C-176 intervention (F = 7.868, *p* < 0.05; F = 262.838, *p* < 0.001). Western blot analysis demonstrated that STING was significantly upregulated in OCYs derived from the locally irradiated and contralateral hind limbs (Figure 6D). Accordingly, the phosphorylation levels of TBK1 were significantly increased in the 12 Gy local IR group, which was accompanied by the increased expression of RANKL and SOST, particularly in the OCYs from the locally irradiated hind limb (Figure 6D). Compared with the vehicle group, when the IR-induced STING activation was blocked by C-176 administration, STING expression and the phosphorylation of TBK1 were inhibited, which was consistent with the inhibition of RANKL and SOST expression (Figure 6D). These results indicated that the STING pathway may participate in the development of IR-induced excessive activation of osteoclastogenesis.

Additionally, the suppression of STING ameliorated the IR-induced biological damage of OCYs. For viability of OCYs, two-way ANOVA demonstrated a significant effect of irradiation (F = 111.905, *p* < 0.001) and a significant effect of C-176 intervention (F = 32.478, *p* < 0.001). Compared with the control group, the viability of primary OCYs derived from the locally irradiated and the contralateral hind limbs was significantly damaged, with observed morphology degenerative changes, such as loss of dendritic structure and cytoplasmic disorder (Figure 6C). However, compared with the locally irradiated vehicle mice group, the degenerative changes were somewhat improved after the STING inhibitor, C-176, intervention (Figure 6C).

Furthermore, 12 Gy local IR can induce multicellular dysfunction of the bone microenvironment. For the volume of BMAs and the ALP expression, two-way ANOVA demonstrated significant effects of irradiation (F = 470.041, *p* < 0.001; F = 61.536, *p* < 0.001). Compared with mice in the control group, the volume of BMAs significantly increased in the locally irradiated (*p* < 0.001) and the contralateral (*p* < 0.01) hind limbs (Figure 7A). This BMA increase was accompanied by a marked decrease in ALP expression in BMSCs (*p* < 0.01; Figure 7C), suggesting that 12 Gy local IR could lead to a significant decrease in the osteogenesis and an elevation in the adipogenesis of bone marrow cells. However, for the volume of BMAs, two-way ANOVA demonstrated no significant effect of C-176 intervention (F = 3.828, *p* > 0.05). When the STING inhibitor, C-176, intervention was conducted, although the inhibitor had only minimal effects on adipogenic differentiation potential, as assessed by the volume of BMAs, C-176 administration could mitigate the IR-induced decline in osteogenesis (*p* < 0.01) (Figure 7C). For the ALP expression, two-way ANOVA demonstrated a significant effect of C-176 intervention (F = 23.844, *p* < 0.001). When C-176 was used to block the STING pathway, the inhibition of the bone marrow osteogenic differentiation ability of mice was successfully mitigated, particularly in the contralateral area that was not directly exposed to IR, and the osteogenic differentiation ability of BMSCs was successfully restored (*p* < 0.01), in which ALP expression was close to the control (Figure 7C).

Finally, the influence of STING pathway intervention on osteoclastogenesis potential was explored. For the TRAP^+^ expression in the cancellous bone area, two-way ANOVA demonstrated a significant effect of irradiation (F = 53.870, *p* < 0.001) and a significant effect of C-176 intervention (F = 52.683, *p* < 0.001). For the TRAP^+^ osteoclasts induced from the BMMs, two-way ANOVA demonstrated a significant effect of irradiation (F = 43.204, *p* < 0.001) and a significant effect of C-176 intervention (F = 49.186, *p* < 0.001). The results demonstrated that a marked promotion of TRAP^+^ expression in the cancellous bone area and the number of TRAP^+^ osteoclasts induced from the BMMs were observed in the locally irradiated and contralateral hind limbs of the 12 Gy locally irradiated group (*p* < 0.05; Figure 7B,D). Nevertheless, after receiving local IR combined with C-176 intervention to block the STING pathway, the TRAP^+^ area of the directly irradiated and the contralateral hind limbs significantly decreased, followed by an inhibited expression of STING, the phosphorylation levels of TBK1 and downstream osteoclastogenesis-stimulated genes, including RANKL and SOST (Figure 6D and Figure 7B). These results indicated that the STING pathway was involved in the IR-induced excessive activation of osteoclastogenesis. In other words, the STING antagonist can block the STING pathway and, thereby, repair IR-induced multicellular biological damage and restore bone homeostasis.

## 4. Discussion

Damage to normal, non-tumor bone tissue following therapeutic irradiation represents a significant concern for patients receiving radiotherapy as part of their cancer treatment. There is a limited understanding of the magnitude and mechanisms behind bone loss associated with radiation exposure. An intricate balance between bone-forming osteoblasts and bone-resorbing osteoclasts plays an important role in maintaining bone integrity and quality through direct cell-to-cell contact or through secretory proteins [46].

In the present study, considering that bone is rich in a heterogeneous population of marrow cells, including hematopoietic precursors and pluripotent mesenchymal stem cells, and bone marrow is radiation-sensitive tissue, the multiple cells in the bone microenvironment from locally irradiated and contralateral hind limbs 7 days after 12 Gy radiation exposure were analyzed. The results suggested that 12 Gy local IR could lead to a significant inhibition in the osteogenic differentiation potential of BMSCs in the locally irradiated and not directly exposed bone tissues. It is, therefore, concluded that BMSCs are likely damaged by radiation exposure, which would likely delay the recovery of osteoblast damage. In our previous research and that of other laboratories, both single 2 Gy total body and 12 Gy local irradiation have been used in animal models and can successfully construct IR-induced bone damage in mice. However, single 2 Gy total body irradiation is only equivalent to the clinical RT routine split single dose and far lower than the total dose in clinical radiotherapy. In addition, total body irradiation may cause a systemic inflammatory response and may even lead to hypogonadism, complicating the interpretation of experimental data related to IR-induced bone microenvironment disruption. In the present study, local 12 Gy irradiation was used to mimic the clinical focal fractionated dose comparable to stereotactic radiotherapy, which may be used to construct a mouse model of IR-induced cellular senescence and multicellular biological damage in bone marrow and may help elucidate the etiological mechanism of cellular senescence-associated bone microenvironment disruption.

It has been demonstrated that the effects of radiation on bone tissues are multifaceted and largely dose-dependent [4]. Low-dose radiation can cause local and distal bone loss, accompanied by a transient increase in osteoclast number or activity, while high-dose ionizing radiation negatively affects osteoclast formation in vitro [47,48,49,50,51]. Osteoclasts, a type of macrophage, play a crucial role in bone remodeling homeostasis, and excessive activation of osteoclasts may cause bone diseases, including osteoporosis and rheumatoid arthritis [52,53]. Therefore, fully elucidating the mechanism of osteoclast differentiation potential is important to understand the etiologies of metabolic bone diseases. In the present study, it was also demonstrated that a transient activation of osteoclasts occurred in the contralateral hind limb, both in the cancellous bone area and in the BMMs extracted from the bone marrow, although a marked reduction in the mature osteoclasts from the locally irradiated hind limb due to the complete suppression of bone marrow and a restraint of the osteoclast progenitors was observed. For the local-irradiated hind limb, the different osteoclast expression between the bone tissue section and BMMs may be mainly caused by the different time schedule. The area of TRAP^+^/bone surface mainly represents the transiently increased OCs that mobilize into bone tissue through the stress response of IR, while OCs from BMMs mainly represent the potential osteoclastgenesis of bone marrow, which is radiosensitive tissue and seriously damaged by IR [18,19,54]. Based on the previous research of our and other laboratories, there appears to be a different cellular mechanism underlying bone damage in different dosages and fractionation modes, especially the osteoclastogenesis potential, which appears as over-activity in the early stages of low-dose IR exposure, while the inhibition of bone-marrow-derived cells was the main manifestation in the simulated clinical RT high-dose radiation exposure [55].

Historically, the drug intervention for bone loss mainly focused on activating the osteoblasts and inhibiting the osteoclasts. However, unidirectional bone-forming activation or bone-resorbing inhibition cannot easily control bone homeostasis accurately, so the efficiency of targeted therapy for only osteogenesis or osteoclastogenesis is not ideal [56,57,58,59]. There is a clear need for further understanding of the communication and mechanisms of multiple types of cells in the bone microenvironment, which may provide more targeted options for the effective intervention of bone homeostasis. OCYs, the most abundant cells in bone tissues, are terminally differentiated cells from mature osteoblasts embedded in mineralized osteoid and are able to modify their surrounding extracellular matrix via specialized molecular remodeling mechanisms. Although osteocyte survival is critical for bone homeostasis, it can be perturbed by age and a variety of stresses, such as ionizing radiation. The present study demonstrated that local radiation can markedly increase osteocyte exfoliation, which is manifested as an increased number of empty lacunae, both in the locally irradiated and in the contralateral bone tissues. Furthermore, the number of primary OCYs harvested from irradiated mice was significantly decreased, along with decreased viability and notable morphology changes in OCYs, particularly in OCYs derived from locally irradiated hind limbs. The effects of radiation on the multicellular modality in the bone microenvironment were examined extensively in the present study. The results suggested that 12 Gy local IR does induce multicellular dysfunction of the bone microenvironment, both in the locally irradiated area and in the contralateral, non-irradiated area, which was manifested by an enhancement in osteocyte exfoliation, activation of osteoclastogenesis, degeneration of osteogenesis and fate conversion of BMSCs into adipocytes. However, this study was designed to observe the short-term effects of bone marrow cell pool as a precursor of bone cells after irradiation, as well as the impact of STING pathway intervention; only the changes in bone tissue and cells 7 days post-irradiation were observed, without observing long-term effects, which is a limitation of this study.

Both low- and high-dose radiation treatment regimens result in bone loss, mainly due to activation of osteoclast-mediated bone resorption and suppression of osteoblast-mediated bone formation [60,61]. Ultimately, DNA damage and cellular apoptosis of the bone lineage cells are also increased. Previous studies suggest that radiation could induce the accumulation of senescent cells by inducing oxidative stress, DNA damage and chromatin disruption, while an inflammatory response does occur in response to radiation and may be the molecular mechanism that causes early activation of osteoclast-mediated bone resorption [60,61,62,63]. Thus, it is unclear whether cells that escape the initial disruption of acute radiation in vivo are metabolically functional or become senescent. In the present study, assessment of senescence-related alterations in cell populations in the bone microenvironment demonstrated that IR can induce the accumulation of senescent cells in bone tissues, as evidenced by increased expression of p16 and p21 in the cortical bone matrix of irradiated mice and the proportion of SA-β-gal^+^ OCYs from locally irradiated mice increasing.

Physiological aging and external stress stimuli, such as drugs and ionizing radiation, can induce cellular senescence, and the long-term accumulation of senescent cells is accompanied by SASP, which triggers a series of degenerative changes and is detrimental for bone tissue regeneration [64,65]. Previous studies have demonstrated that the secretion of aging-related inflammatory cytokines is closely related to senile diseases, such as osteoporosis, senile osteoarthritis and other bone metabolism diseases [66]. Studies have confirmed that a variety of SASP components, such as IL-1β, IL-6, TNF-α and MMP-13, which are inflammatory factors, acted on bone cells through paracrine secretion, and they manifested as effects on inhibiting the bone formation or/and promoting bone resorption [15,16]. Among them, secretory inflammatory factors are highly correlated with the regulation of bone metabolism, particularly the chemotaxis and differentiation maturation of osteoclast precursors [15]. In the present study, semi-quantitative microarray was performed to analyze the differential expression changes in typical components of SASP in the serum of experimental mice. The results demonstrated that the expression of SASP factors was significantly altered when mice received 12 Gy local IR. Among the 111 cytokines detected, the expression levels of 56 SASP factors were significantly upregulated in the irradiated mice, including the interleukin family, chemokines, growth factors, adipokines, extracellular matrix proteins and serum-associated inflammatory factors. It is known that a complex combination of SASP factors can promote osteoclastogenesis via direct and paracrine regulatory forms [16]. Exploring alternative mechanisms in the present study, it was demonstrated that local radiation causes bone aging accompanied by altered inflammatory cytokines as part of SASP and upregulated expression of multiple factors involved in the process of osteoclastogenesis.

While much progress has been achieved in understanding cellular senescence and SASP regulation pathways, understanding the intrinsic links between SASP pathways and the molecule mechanisms by which SASP adjusts bone homeostasis may yet reveal deeper complexities. It has been demonstrated that STING can regulate a variety of inflammatory secretory phenotypes through the STING-P-TBK1 pathway [32,33]. The expression and role of the STING pathway have previously been reported in osteoclastic precursors [67,68]. However, to the best of our knowledge, there is still a lack of research evidence for the role of the STING pathway in the regulation of osteoclastogenesis by paracrine facilitation due to senescent osteogenic stem cells. 

In the present study, the potential role of STING on the progression of inflammatory cytokine production and osteoclastogenesis was investigated using a locally irradiated mouse model and osteocyte-like MLO-Y4 cells in vitro model. It was demonstrated that the STING pathway was activated in both of these models. In the present study, the expression of STING in primary OCYs harvested from locally irradiated or contralateral hind limbs was elevated significantly. Accordingly, the phosphorylation levels of TBK1, which is the direct downstream signaling protein of STING, were also significantly increased. Consistent with the activation of the STING-P-TBK1 pathway, the expression levels of the relevant functional proteins secreted by OCYs were also altered, including a markedly increased expression of RANKL and SOST and decreased expression of OPG in OCYs derived from the locally irradiated and contralateral hind limbs. Bone homeostasis is maintained by coordinated cycles of bone resorption and formation. The receptor activator of NF-κB ligand (RANKL) binds RANK on the surface of osteoclast precursors to induce osteoclast-associated gene expression and trigger osteoclastogenesis [69]. Cumulative evidence indicates that OCYs are the major source of RANKL during the bone remodeling process, and the expression of RANKL of OCYs can be used as a direct indicator of the regulation of osteoclast production [16]. Hence, the results of the present study demonstrated that ionizing radiation can alter the expression of related functional proteins secreted by OCYs by activating the STING-P-TBK1 pathway. 

Knockdown of STING expression with STING-siRNA in cultured OCY-like MLO-Y4 cells was followed by a corresponding inhibition of the IR-induced activation of RANKL protein expression, along with a downregulation of P-TBK1 protein expression, confirming the crucial role of STING in the secretion of bone-resorption-related protein functional RANKL. Similarly, the mRNA expression levels of inflammatory cytokines, such as IL-1α, IL-6 and NF-κB, which are the important modulators of osteoclastogenesis, were significantly upregulated in irradiated OCYs. A successful restoration of IR-induced activation of the IL-1α and NF-kB genes was detected after STING-siRNA knocked down the STING pathway.

Furthermore, the co-culturing of RAW264.7 cells with the CM of IR-induced senescent MLO-Y4 cells, with or without STING-siRNA intervention, demonstrated that blocking the STING pathway reversed the over-activation of osteoclastogenesis. These findings suggested that IR could activate the STING-P-TBK1 pathway, allowing for altering the secretion of OCYs-derived related functional proteins and regulating osteoclast differentiation. Using STING-siRNA-targeted intervention to block the STING pathway can effectively inhibit IR-induced paracrine changes in bone cells and their regulation of hyperactive osteoclast differentiation. In agreement with the present study, the association of STING activation in bone metabolism has been reported, and attenuation of osteoclastogenesis can be achieved by inhibiting STING-dependent NF-κB signaling [68]. In this study, it was the activation of the STING signaling pathway caused by IR-induced senescent OCYs that paracrine-regulated osteoclastogenesis and bone homeostasis by the secretion of inflammatory factors. This evidence indicated that STING knockdown regulated osteoclastogenesis via inhibiting the phosphorylation of signal transducer and secretion of inflammatory cytokines. This may open up new opportunities for the treatment of IR-induced bone homeostasis disruption through STING inhibition via modulation of osteocyte–osteoclast crosstalk.

In order to support the merit of conducting a STING intervention for IR-induced bone metabolism imbalance, an interventional block of the STING pathway via administration of a STING inhibitor, C-176, in mice with 12 Gy local IR exposure was investigated. Compared with the IR-induced vehicle mice group, C-176 administration decreased the levels of STING and inhibited the phosphorylation of TBK1. This was consistent with the reversion of enhanced downstream osteoclastogenesis-stimulated genes, including RANKL and SOST, along with a mitigation of enhanced osteoclastogenesis potential, suggesting that the STING pathway participates in the development of IR-induced excessive activation of osteoclastogenesis. Additionally, C-176 intervention ameliorated the IR-induced biological damage of multicellular dysfunction in the bone microenvironment. Given these findings, the degenerative changes in primary OCYs were somewhat improved after C-176 intervention. Meanwhile, C-176 administration successfully restored IR-induced decline of osteogenesis but only exhibited minimal effects on adipogenic differentiation potential. Cumulative evidence in the present study indicated that the STING-P-TBK1 pathway is an important signaling pathway that affects the multicellular biological function and functional proteins in the bone microenvironment, thereby affecting paracrine regulation in bone homeostasis.

Findings from the present study indicated that ionizing radiation can activate the STING pathway in OCYs, therewith enhancing the expression of inflammatory cytokines and regulating the function of bone microenvironment cells, in particular, activating osteoclastogenesis via the paracrine pathway. Selective antagonist treatment can be administered to block the STING pathway and, thereby, repair IR-induced multicellular biological damage. Although the STING inhibitor had only minimal effects on adipogenic differentiation potential, it mitigated IR-induced excessive activation of osteoclastogenesis and restrained osteogenesis, thereby restoring the balance of bone remodeling. These results suggest that the STING pathway may be an important signaling pathway for the paracrine regulation of OCYs and in the imbalance of bone metabolism caused by ionizing radiation, due to its bidirectional secretory regulation of bone resorption and bone formation.

In the present study, it was concluded that 12 Gy local IR does induce multicellular dysfunction of the bone microenvironment, both in the locally irradiated area and in the contralateral, non-irradiated area. This was accompanied by bone aging and altered SASP secretion. In this study, male 6-week-old BALB/c mice were adopted and successfully established IR-induced bone homeostasis disruption to observe the multicellular senescence in the bone microenvironment and its heterogeneous SASP. Six-week-old mice have reached sexual maturity, which is the best age for animal experiments, including IR-induced bone loss as a model for bone aging. However, it is in older survivors where the deleterious synergy between aging and radiation exposure conspires to promote tissue deterioration and dysfunction, which then negatively impacts their quality of life. Therefore, consideration of the crosstalk between senescence and irradiation may be a practical experimental design. 

Another limitation of the present study is the radiation dosage and fractionation mode, which could affect the pathological process of bone damage after RT. In a previous study, our results showed that compared with the bone injury caused by single 2 Gy local radiation, the bone damage after three 8 Gy doses of local radiation appears earlier, lasts for a longer period of time and is not easily reversed. In the present study, single 12 Gy local irradiation was conducted to simulate the clinical SBRT high-dose model, instead of three fractionated doses of 8 Gy local irradiation model, which could truly simulate the clinical RT dose and fractionated mode but lacks practical applicability due to numerous administrations of anesthetic [55]. However, the result of this study showed IR-induced bone microenvironment disruption, which was similar to the results of the previous low-dose study in our laboratory. In a previous experimental study, total body irradiation of mice using a lower dose of 2 Gy showed the decreased osteogenic capacity of irradiated mice BMSCs, and increased the number of cortical bone cavities, suggesting that lower doses of radiation adversely affect the homeostasis of the bone microenvironment. 

Compared with previous studies in our laboratory, this study further explored the molecular mechanisms that regulate the bone microenvironment. IR can activate the STING-P-TBK1 pathway, along with an alteration in OCY-derived inflammatory cytokines and osteoclastogenesis potential, which could be mitigated by blocking the STING pathway, thereby restoring the balance of bone remodeling. This new finding may provide an important basis for elucidating the complex pathological mechanisms of osteoporosis and provide promising drugs for clinical osteoporosis treatment. However, the present study did not explore in depth the role of the STING-TBK1 signaling pathway in the fracture effect induced by radiotherapy, and the experimental design lacked a direct display of bone damage in terms of bone volume and bone density. In addition, more experimental data are needed for exploring the regulation of specific molecules downstream of the STING pathway. Targeting the STING pathway and its regulation of inflammatory cytokines is expected to be an effective intervention strategy to protect against IR-induced bone homeostasis imbalance and degenerative injury.

## 5. Conclusions

The STING-P-TBK1 signaling pathway plays a crucial role in the regulation of the secretion of inflammatory cytokines and osteoclastogenesis potential in IR-induced bone microenvironment disruption. The selective STING antagonist can be used to intervene to block the STING pathway and, thereby, repair IR-induced multicellular biological damage and successfully mitigate IR-induced activation of osteoclastogenesis and restraint of osteogenesis. Targeting the STING pathway may be an effective intervention strategy to protect against radiotherapy-associated bone turnover imbalance.

## Figures and Tables

**Figure 1 medicina-59-01316-f001:**
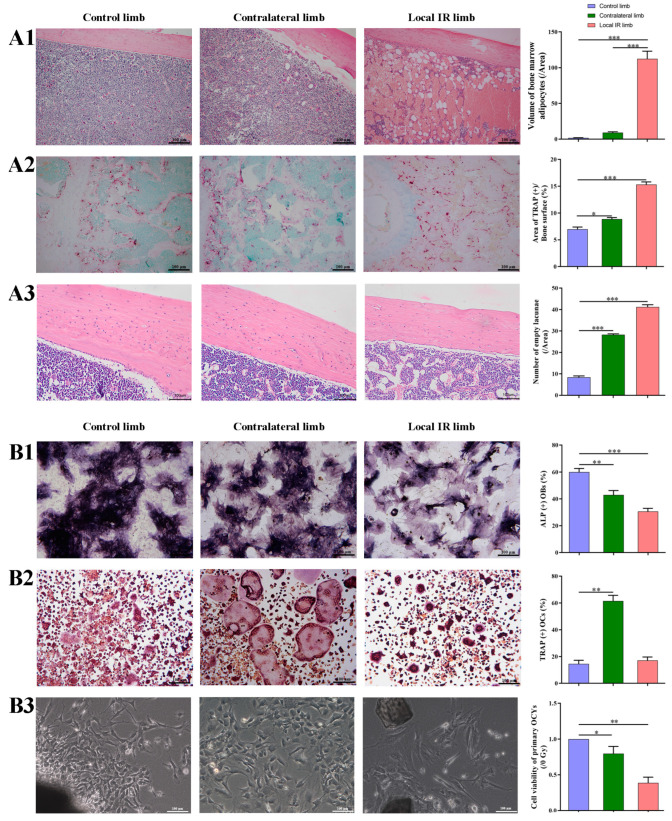
IR-induced multicellular dysfunction of bone microenvironment in a mouse model of bone degenerative injury. (**A1**) The representative images (HE staining) of adipogenic differentiation potential and the quantitative analysis of the volume of BMAs in tissue section. (**A2**) The representative images of osteoclastogenesis potential and quantitative analysis of TRAP staining of distal femur trabecular bone. (**A3**) The representative images of biological damage of OCYs and quantitative analysis of empty bone lacuna in femoral bone tissue sections by HE staining. (**B1**) The representative images of osteogenic differentiation potential and quantitative analysis of ALP staining of BMSCs harvested from the bone marrow. (**B2**) The representative images of osteoclastogenesis potential and quantitative analysis of TRAP staining of BMMs harvested from the bone marrow of the mice and induced with RANKL stimulation. (**B3**) The biological damage of OCYs. The morphology of primary OCYs harvested from the experimental mice, and the quantitative analysis of cell viability of primary OCYs by the CCK-8. Data are expressed as mean ± SD (* *p* < 0.05; ** *p* < 0.01; *** *p* < 0.001 vs. control), *n* = 6. Scale bar, 100 µm. Magnification, ×100. The hind limb of the control group: Control limb; the left hind limb of the local irradiation group: Local IR limb; the right hind limb of the local irradiation group: Contralateral limb. BMAs, bone marrow adipocytes; CCK-8, Cell Counting Kit-8 assay.

**Figure 2 medicina-59-01316-f002:**
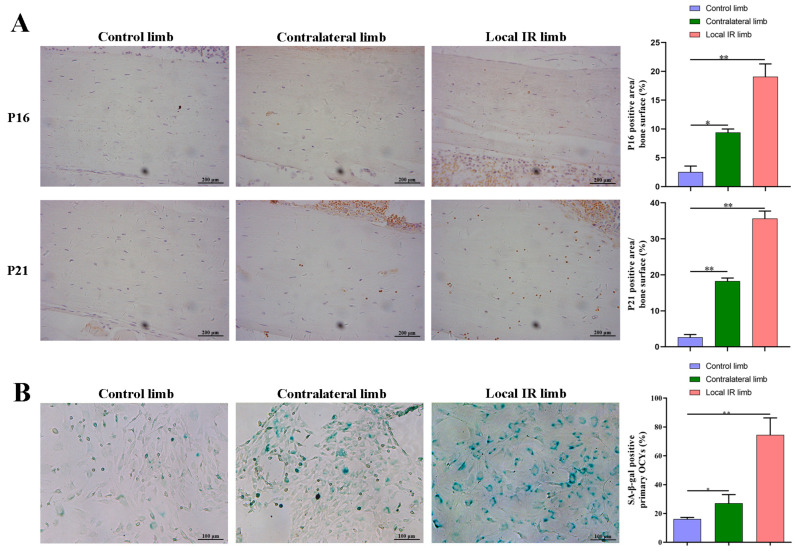
IR-induced bone aging in a mouse model of bone-degenerative injury. (**A**) The representative images and quantitative analysis of the expression of the senescence-related proteins, p21 and p16 of bone tissue sections by immunohistochemistry staining. Scale bar, 200 µm. Magnification, ×200. (**B**) The representative images of SA-β-ga staining in primary OCYs harvested from the experimental mice, 7 days post-irradiation, and quantitative analysis of the SA-β-gal^+^ OCYs. Scale bar, 100 µm. Magnification, ×100. Data are expressed as mean ± SD (* *p* < 0.05; ** *p* < 0.01;vs. control), *n* = 6.

**Figure 3 medicina-59-01316-f003:**
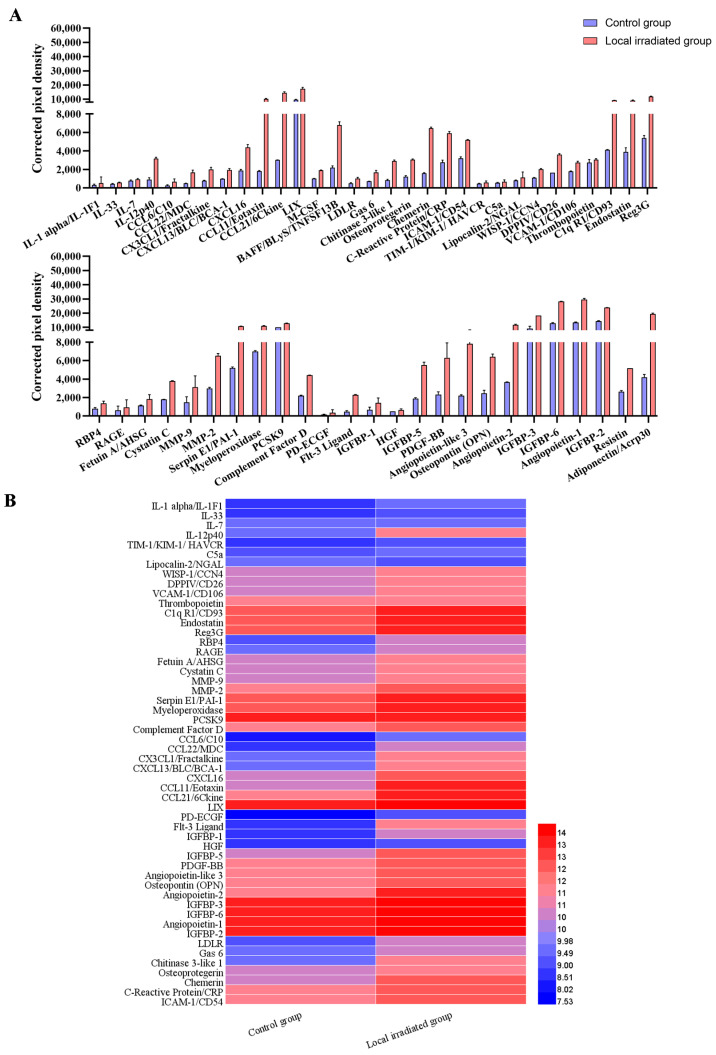
IR-induced expression changes in the secretion SASP components in a mouse model of bone degenerative injury. (**A**) The SASP composition in mice serum post-irradiation was detected by Mouse Cytokine microarray membrane analysis. Quantified expression levels of the secretory cytokines, expressed as corrected the pixel density, *n* = 2. The criteria for defining up- or downregulation was greater or less than 2 SD by the mean value of corrected pixel density in the control group. (**B**) Heatmap representation of cytokines based on antibody microarray results. Data are expressed as mean ± SD.

**Figure 4 medicina-59-01316-f004:**
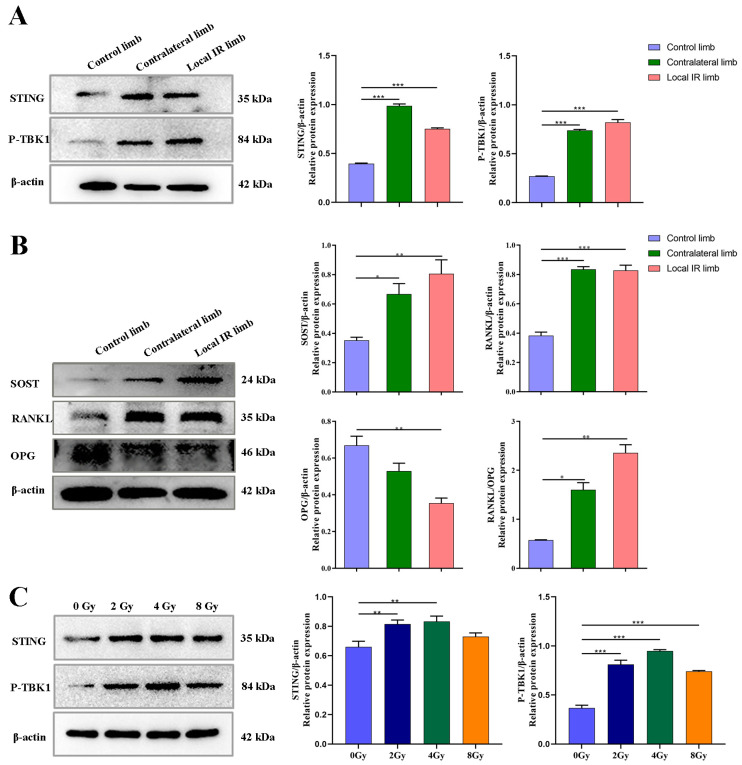
Radiation activates the STING-P-TBK1 pathway and alters the inflammatory secretion. (**A**) The expression levels of pathway-related proteins STING and P-TBK1 of primary OCYs harvested from the experimental mice by western blot analysis. (**B**) The expression levels of the bone resorption-related functional proteins SOST, RANKL, and OPG of primary OCYs by western blot analysis. The ratio of RANKL/OPG expression levels of primary OCYs. (**C**) An in vitro OCY model of irradiation, the expression levels of the pathway-related proteins STING and P-TBK1 3 days post-irradiation by Western blot analysis. Data are expressed as mean ± SD (* *p* < 0.05; ** *p* < 0.01; *** *p* < 0.001 vs. control), *n* = 3.

**Figure 5 medicina-59-01316-f005:**
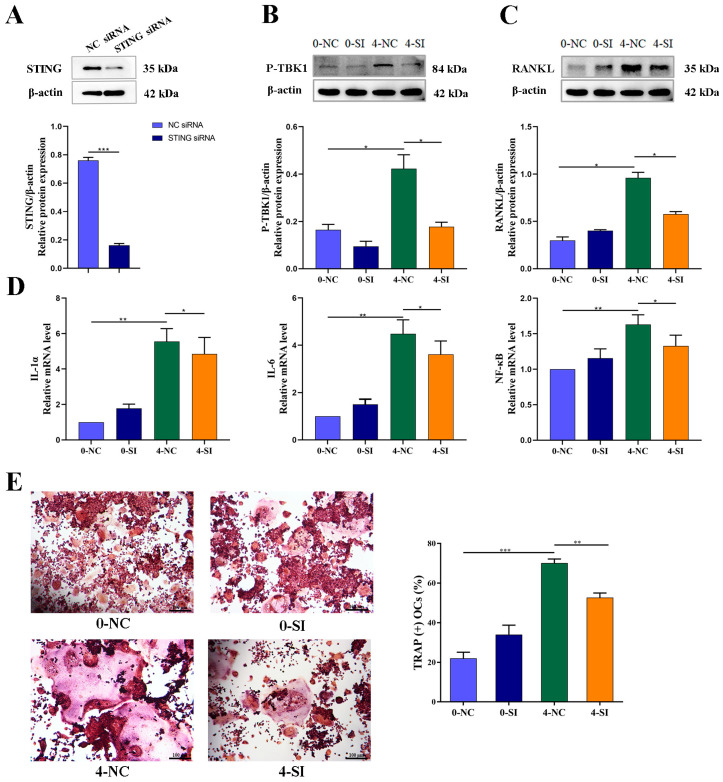
STING-P-TBK1 pathway blocking by siRNA alters inflammatory cytokines secretion and osteoclastogenic activity in an in vitro osteocyte model of irradiation. (**A**) The knockdown efficiency of STING was verified by western blot analyses. (**B**) After the STING-siRNA knockdown, the protein expression of P-TBK1 in MLO-Y4 cells was evaluated 3 days post-irradiation by western blot analysis. (**C**) Effect of STING-targeted siRNA intervention on the expression of bone resorption-related protein RANKL. (**D**) Effect of STING-targeted siRNA intervention on the mRNA expressions of inflammatory cytokines including IL-1α, IL-6, and NF-κB. (**E**) STING-targeted siRNA intervention on the paracrine regulation of osteoclastogenesis by IR-induced inflammatory cytokine from OCYs. The RAW264.7 cells were co-cultured with the conditioned medium (CM) of irradiated MLO-Y4 cells following STING-siRNA or NC siRNA intervention. The percentage of TRAP^+^ OCs was quantified (scale bar, 100 µm; magnification, ×100). Data are expressed as mean ± SD (* *p* < 0.05; ** *p* < 0.01; *** *p* < 0.001 vs. control), *n* = 3. NC siRNA: non-STING targeted siRNA interference; STING siRNA: STING targeted siRNA interference. The CM of MLO-Y4 cells was collected at 3 days after 0 Gy or 4 Gy irradiation with STING siRNA or NC siRNA intervention. CM-0 Gy-NC: non-irradiation with NC siRNA intervention; CM-4 Gy-NC: 4 Gy γ-radiation with NC siRNA intervention; CM-0 Gy-SI: non-irradiation with STING-siRNA intervention; CM-4 Gy-SI: 4 Gy γ-radiation with STING-siRNA intervention.

**Figure 6 medicina-59-01316-f006:**
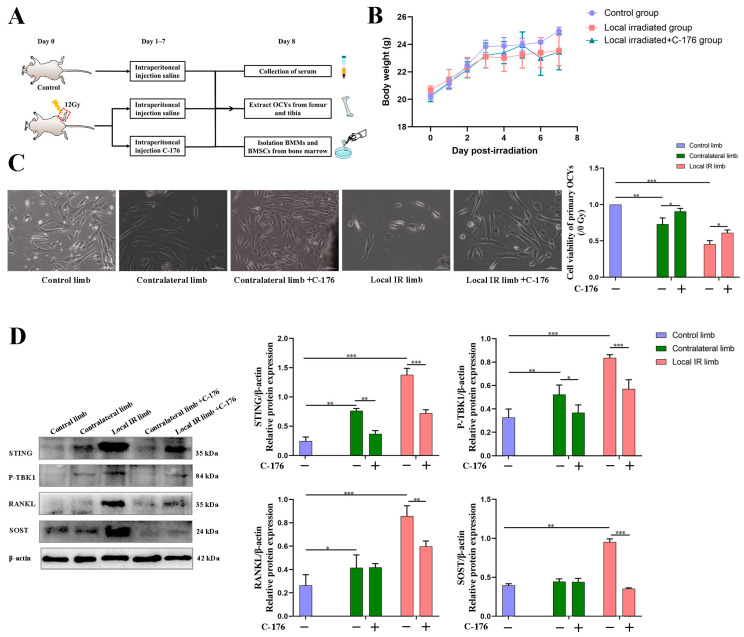
C-176 blocks the STING pathway and ameliorates the excessive inflammatory secretion in a mouse model of bone degeneration. (**A**) Pattern diagram of the experiment. Control group: no irradiation; local irradiation group: pre-treated with 12 Gy local irradiation; local irradiation group + C-176: pre-treated with 12 Gy local irradiation and given intraperitoneal injections of STING small molecule inhibitor C-176. (**B**) Effect of 12 Gy local irradiation and C-176 intervention on body weight in mice, *n* = 12. (**C**) Changes of morphology and cell viability of primary OCYs harvested from the experimental mice 7 days post-irradiation (scale bar, 100 µm; magnification, ×100. Cell viability of OCYs was quantified using the CCK-8, *n* = 6. (**D**) Western blot analysis of the expressions of the pathway-related proteins STING and P-TBK1, and the bone resorption-related proteins SOST and RANKL of primary OCYs, *n* = 6. Data are expressed as mean ± SD (* *p* < 0.05; ** *p* < 0.01; *** *p* < 0.001 vs. control).

**Figure 7 medicina-59-01316-f007:**
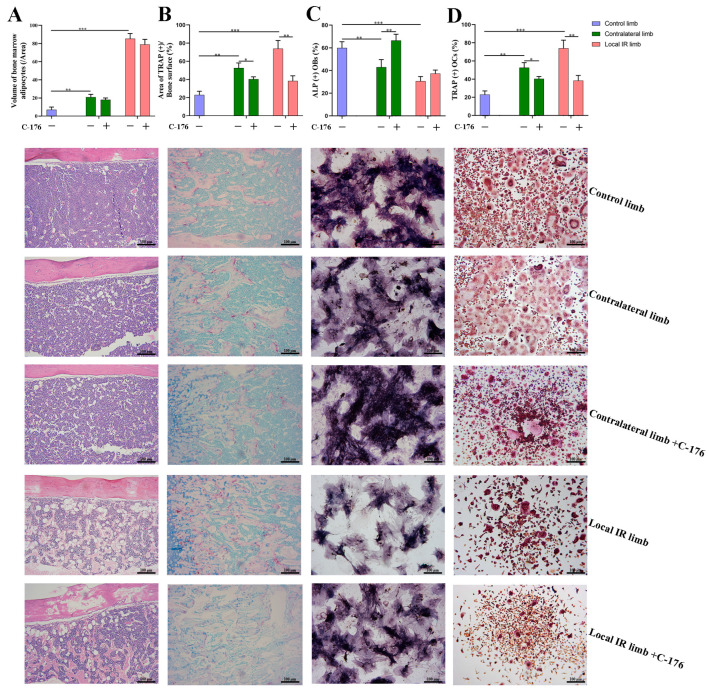
C-176 ameliorates the IR-induced bone turnover imbalance. (**A**) The quantitative analysis and the representative images (HE staining) of the volume of BMAs in tissue section, showed the intervention of C-176 on the IR-induced active adipogenic differentiation potential. (**B**) The quantitative analysis and representative images of TRAP staining of distal femur trabecular bone showed the intervention of C-176 on the IR-induced active osteoclastogenesis. (**C**) The quantitative analysis and representative images of ALP staining of BMSCs harvested from the experimental mice, showed the intervention of C-176 on the IR-induced degenerative osteogenesis. (**D**) The quantitative analysis and representative images of TRAP^+^ cells of BMMs harvested from the experimental mice, showed the intervention of C-176 on the IR-induced excessive osteoclastogenesis potential. Data are expressed as mean ± SD (* *p* < 0.05; ** *p* < 0.01; *** *p* < 0.001 vs. control), *n* = 5. Scale bar, 100 µm. Magnification, ×100.

**Table 1 medicina-59-01316-t001:** RT-qPCR reaction system.

System Composition	Volume (μL)
PowerUp SYBR-Green Master Mix (2×)	5 μL
PCR Forward Primer (10 μM)	0.5 μL
PCR Reverse Primer (10 μM)	0.5 μL
template cDNA	1 μL
ddH_2_O	3 μL
total capacity	10 μL

**Table 2 medicina-59-01316-t002:** Primer sequence for RT-qPCR.

Target Genes	Forward (5’-3’)	Reverse (5’-3’)
NF-κB	TGCGATTCCGCTATAAATGCG	ACAAGTTCATGTGGATGAGGC
IL-1α	CTGAAGAAGAGACGGCTGAGT	CTGGTAGGTGTAAGGTGCTGAT
IL-6	ATGAACAACGATGATGCACTTG	GGTACTCCAGAAGACCAGAGG
GAPDH	GGAGTCTACTGGTGTCTTC	TCATCATACTTGGCAGGTT

Note. GAPDH: glyceraldehyde-3-phosphate dehydrogenase.

**Table 3 medicina-59-01316-t003:** Changes in SASP patterns in a local radiation-induced senescence animal model.

Category	SASP Factors	Change Pattern *
Interleukins (IL)	IL- 1α, IL-7, IL-12p40, IL-33	↑
	IL-11	-
	IL-1β, IL-2, IL-3, IL-4, IL-5, IL-6, IL-10, IL-13, IL-15, IL-17A, IL-22, IL-23, IL-27p28, IL-28	↓
Chemokines (CXCL, CCL)	CCL6/C10, CCL11/Eotaxin, CCL21/6Ckine, CXCL16, CCL22/MDC, CXCL13/BLC/BCA-1, CX3CL1/Fractalkine, LIX	↑
	CCL2/JE/MCP-1, CCL3/CCL4 MIP-1 alpha/beta, CCL5/RANTES, CCL12/MCP-5, CCL17/TARC, CCL19/MIP-3 beta, CCL20/MIP-3 alpha, CXCL11/I-TAC, CXCL1/KC, CXCL2/MIP-2, CXCL9/MIG, CXCL10/IP-10	↓
Other inflammatory factor	BAFF/BLyS/TNFSF13B	↑
	GDF-15, GM-CSF, IFN-gamma, G-CSF, TNF-alpha, Coagulation factor III/tissue factor	↓
Growth factors and regulators	Angiopoietin-1, Angiopoietin-2, Angiopoietin-like 3, Flt-3 ligand, IGFBP-1, IGFBP-2, IGFBP -3, IGFBP-5, IGFBP-6, Osteopontin (OPN), PD-ECGF/Thymidine phosphorylase, PDGF-BB	↑
	Fetuin A/AHSG, VEGF, HGF	-
	Amphiregulin, EGF, FGF acidic, FGF-21, Proliferin	↓
Proteases and regulators	M-CSF, MMP-2, MMP-9, Myeloperoxidase, RAGE, Complement factor D, RBP4, Fetuin A/AHSG, Cystatin C, Proprotein convertase 9/PCSK9, Serpin E1/PAI-1	↑
	MMP-3	-
	Serpin F1/PEDF	↓
Soluble or shed receptors or ligands	ICAM-1/CD54, Osteoprotegerin/TNFRSF11B, Chemerin, Chitinase 3-like 1, Gas6, LDL R, C-reactive protein/CRP	↑
	LIF	↓
Adipokines	Adiponectin/Acrp30, Resistin	↑
	Leptin	-
	Pref-1/DLK-1/FA1	↓
Extracellular matrix protein	C1q R1/CD93, DPPIV/CD26, Endostatin, Lipocalin-2/NGAL, Reg3G, VCAM-1/CD106, WISP-1/CCN4	↑
	Periostin/OSF-2, TIM-1/KIM-1/HAVCR, Complement component C5/C5a, Thrombopoietin, DKK-1	-
	CD160, CD14, CD40/TNFRSF5, Endoglin/CD105, Pentraxin 2/SAP, Pentraxin 3/TSG-14, P-selectin/CD62P, Coagulation factor III/tissue factor, E-selectin/CD62E	↓

***** When the value of corrected pixel density was 2 standard deviations higher or lower than the control group, the index was defined as up or down regulation. ↑: Up regulation. -: No significant change. ↓: Down regulation.

## Data Availability

The datasets generated and analyzed in the present study are all included in the published article.

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
