# Peer review of "Radiation Induces Bone Microenvironment Disruption by Activating the STING-TBK1 Pathway"

_medicina, 2023, doi:10.3390/medicina59071316_

Round 1
Reviewer 1 Report
The manuscript by Yuyang Wang and collaborators presents a very interesting and well conducted study investigating the molecular mechanisms of bone damage induced by radiations.
Several points need to be addressed before publication:
-The reporting of the results and the numbering in Figure 1 both cause confusion. Authors should distinct histological and cell analyses
- The presence or absence of statistically significant changes in SASP patterns should be clearly state. Moreover, table 3 is quite misleading. Although the authors want to report an increase/decrease trend, the expression of many markers is absolutely not altered (as you can guess from the figure). The table should be modified. In addition, the most relevant alterated markers should be validated by an alternative techniques
-The levels of RANKL and OPG should be also investigated in conditioned media. Moreover, OPG/RANKL ratio should be also calculated and discussed
Author Response
Thanks for your encouragement and comments on our paper (medicina-2467031). These comments are valuable for revising our paper and our subsequent research work. We have discussed the comments carefully and revised the paper which we hope meet with approval.
Point 1: The reporting of the results and the numbering in Figure 1 both cause confusion. Authors should distinct histological and cell analyses.
Response 1: We have rearranged Figure 1 according to histological and cellular level results.
Point 2: The presence or absence of statistically significant changes in SASP patterns should be clearly state. Moreover, table 3 is quite misleading. Although the authors want to report an increase/decrease trend, the expression of many markers is absolutely not altered (as you can guess from the figure). The table should be modified. In addition, the most relevant alterated markers should be validated by an alternative techniques.
Response 2: Sorry for ambiguity in Table 3. We have optimized the description of Table 3, classifying factors with indicator values that did not change by more than 2 standard deviations from the control group as no significant change.
Secondly, in this study, an antibody microarray approach was used to examine the expression of 111 SASP cytokines, and the mRNA expression levels were validated for key SASP factors involved in the regulation of osteoclastogenesis. In future, we hope to validate ulteriorly the most relevant alterated markers one by one.
Point 3: The levels of RANKL and OPG should be also investigated in conditioned media. Moreover, OPG/RANKL ratio should be also calculated and discussed.
Response 3: So sorry for the lack of analysis of RANKL and OPG levels in conditioned media. Alternatively, in accordance with the research hypothesis and the available results, the ratio of RANKL/OPG was calculated and discussed according to the reviewer's suggestion, see Figure 4B.
Please see the attachment.

Reviewer 2 Report
The authors hypothesis the physiological as well as pathological processes during IR–induced bone homeostasis deterioration in order to establish a broad understanding of the biomechanical as well as molecular mechanisms involved in senescent microenvironment and bone homeostasis deterioration. In addition, evidence for the use of STING inhibitors as potential therapeutic agents targeting bone resorption in IR-induced bone microenvironment disruption will be discussed. We highlight the diversity of the senescent cells in the microenvironment of bone, as well as the mechanisms by which these senescent cells are involved in musculoskeletal diseases, such as IR-induced bone homeostasis deterioration. In conclusion, the STING-P-TBK1 signaling pathway plays a crucial role in the regulation of the 951 secretion of inflammatory cytokines and osteoclastogenesis potential in IR-induced bone microenvironment disruption.
The introduction is well written , with adequate bibliographic references . However, it is excessively long and should be reduced The methodology is complete, widely described, which would allow the study to be carried out by another research group. The results are meticulously described following a physiopathological scheme that allows us to test the proposed hypothesis. The discussion is correct, adapting to the results obtained.
Author Response
Point 1: The authors hypothesis the physiological as well as pathological processes during IR–induced bone homeostasis deterioration in order to establish a broad understanding of the biomechanical as well as molecular mechanisms involved in senescent microenvironment and bone homeostasis deterioration. In addition, evidence for the use of STING inhibitors as potential therapeutic agents targeting bone resorption in IR-induced bone microenvironment disruption will be discussed. We highlight the diversity of the senescent cells in the microenvironment of bone, as well as the mechanisms by which these senescent cells are involved in musculoskeletal diseases, such as IR-induced bone homeostasis deterioration. In conclusion, the STING-P-TBK1 signaling pathway plays a crucial role in the regulation of the 951 secretion of inflammatory cytokines and osteoclastogenesis potential in IR-induced bone microenvironment disruption.
The introduction is well written, with adequate bibliographic references. However, it is excessively long and should be reduced The methodology is complete, widely described, which would allow the study to be carried out by another research group. The results are meticulously described following a physiopathological scheme that allows us to test the proposed hypothesis. The discussion is correct, adapting to the results obtained.
Response 1: Thanks for your attention and giving useful comments. As recommended, we have shortened and simplified the introduction section. See introduction section.
Please see the attachment.

Round 2
Reviewer 1 Report
Authors addressed all requests sufficiently. Now, the manuscript is ready for publication.